# Carnitine Protects against MPP^+^-Induced Neurotoxicity and Inflammation by Promoting Primary Ciliogenesis in SH-SY5Y Cells

**DOI:** 10.3390/cells11172722

**Published:** 2022-09-01

**Authors:** Ji-Eun Bae, Joon Bum Kim, Doo Sin Jo, Na Yeon Park, Yong Hwan Kim, Ha Jung Lee, Seong Hyun Kim, So Hyun Kim, Mikyung Son, Pansoo Kim, Hong-Yeoul Ryu, Won Ha Lee, Zae Young Ryoo, Hyun-Shik Lee, Yong-Keun Jung, Dong-Hyung Cho

**Affiliations:** 1Brain Science and Engineering Institute, Kyungpook National University, Daegu 41566, Korea; 2School of Life Sciences, BK21 FOUR KNU Creative BioResearch Group, Kyungpook National University, Daegu 41566, Korea; 3ORGASIS Corp. 260, Changyong-daero, Yeongtong-gu, Suwon 16229, Korea; 4Biocenter, Gyeonggido Business and Science Accelerator, Suwon 16229, Korea; 5School of Biological Sciences, Seoul National University, Seoul 08826, Korea

**Keywords:** primary cilia, L-carnitine, acetyl-L-carnitine, MPP^+^, Mitochondria, SH-SY5Y cells

## Abstract

Primary cilia help to maintain cellular homeostasis by sensing conditions in the extracellular environment, including growth factors, nutrients, and hormones that are involved in various signaling pathways. Recently, we have shown that enhanced primary ciliogenesis in dopamine neurons promotes neuronal survival in a Parkinson’s disease model. Moreover, we performed fecal metabolite screening in order to identify several candidates for improving primary ciliogenesis, including L-carnitine and acetyl-L-carnitine. However, the role of carnitine in primary ciliogenesis has remained unclear. In addition, the relationship between primary cilia and neurodegenerative diseases has remained unclear. In this study, we have evaluated the effects of carnitine on primary ciliogenesis in 1-methyl-4-phenylpyridinium ion (MPP^+^)-treated cells. We found that both L-carnitine and acetyl-L-carnitine promoted primary ciliogenesis in SH-SY5Y cells. In addition, the enhancement of ciliogenesis by carnitine suppressed MPP^+^-induced mitochondrial reactive oxygen species overproduction and mitochondrial fragmentation in SH-SY5Y cells. Moreover, carnitine inhibited the production of pro-inflammatory cytokines in MPP^+^-treated SH-SY5Y cells. Taken together, our findings suggest that enhanced ciliogenesis regulates MPP^+^-induced neurotoxicity and inflammation.

## 1. Introduction

Primary cilia are dynamically regulated, highly conserved organelles that project from the surfaces of most human cells [1,2]. They play critical roles in sensing the extracellular environment, including the presence of growth factors, nutrients, and hormones that are involved in the developmental and homeostatic signaling pathways [2,3]. Because cilia regulate multiple signaling pathways, the cilia membranes harbor various signaling receptors, ion channels, and their components, including sonic hedgehog (Shh) and platelet-derived growth factor (PDGF) receptors [2,3,4]. Moreover, the dysregulation of the primary cilia function that is caused by the loss of ciliary proteins induces cell death, whereas enhanced primary ciliogenesis reduces cell death after ischemic injury [4]. Furthermore, the disruption of the cilia regulatory functions seems to underlie a diverse spectrum of disorders, which are known as primary ciliopathies. Primary ciliopathies include numerous, seemingly unrelated, developmental syndromes that can manifest in the retina, the kidney, the liver, the pancreas, the skeletal system, and the brain [5]. Primary cilia are usually present in quiescent or differentiated cells. The cilium assembly proceeds through a series of orchestrated stages, resulting in the remodeling of the maternal centriole [6]. In contrast, cilium disassembly requires the destabilization and depolymerization of axonemal microtubules [6]. Many ciliary proteins have locations and functions in cilium. The intraflagellar transport (IFT) system regulates the assembly and the maintenance of the cilia by mediating protein trafficking [7,8]. Multiple IFT-A and IFT-B complexes function in both retrograde and anterograde IFT. Moreover, several ciliary proteins, such as ADP-ribosylation factor-like protein 13B (ARL13B) and Smoothened (Smo), are commonly used as markers for the monitoring of the primary cilia [9,10]. 

Parkinson’s disease (PD) is a common neurodegenerative disorder that is characterized by a progressive loss of dopaminergic (DA) neurons. The defective mitochondrial function and the resulting increase in oxidative stress, neuroinflammation, and microglial activation reportedly play important roles in PD pathogenesis [11]. Interestingly, the mitochondrial neurotoxin 1-methyl-4-phenyl-1,2,3,6-tetrahydropyridine (MPTP) produces most of the biochemical and pathological hallmarks of PD. MPTP produces a parkinsonian syndrome after it is converted to 1-methyl-4-phenylpyridinium ion (MPP^+^) by type B monoamine oxidase. The DA reuptake system allows MPP^+^ to accumulate in DA neurons, which impairs the mitochondrial respiration by inhibiting complex I [12]. Moreover, mutations in the *LRRK2* gene are the predominant cause of familial PD. A recent study found that cholinergic interneurons of the dorsal striatum lost cilia in mice that carried a PD-associated kinase-activating mutation in the *LRRK2* gene [13]. In addition, we have shown that enhanced primary ciliogenesis in dopamine neurons promotes autophagy and neuronal survival in an MPTP-induced PD model [14].

Recently, we have developed a cell-based screening system using retinal pigment epithelium (RPE) cells and have screened a fecal metabolite library in order to identify several putative candidates for regulating primary ciliogenesis, including L-carnitine and acetyl-L-carnitine [15]. However, the role of carnitine in primary ciliogenesis has remained unclear. In this study, we have evaluated the effect of carnitine on the primary ciliogenesis in MPP^+^-treated cells. We have found that L-carnitine and acetyl-L-carnitine increased the primary ciliogenesis. Furthermore, the enhanced ciliogenesis with carnitine suppressed MPP^+^-induced mitochondrial reactive oxygen species (ROS) overproduction and mitochondrial fragmentation. In addition, carnitines inhibited the production of pro-inflammatory cytokines in the MPP^+^-treated SH-SY5Y cells.

## 2. Materials and Methods

### 2.1. Cell Culture and Stable Cell Lines

In this study, we used human telomerase-immortalized RPE cells and RPE/Smo-GFP cells that stably expressed Smo-GFP proteins, as previously described [15]. The SH-SY5Y neuroblastoma cells were obtained from the American Type Culture Collection (ATCC, Manassas, VA, USA). The SH-SY5Y cells were transfected with pmito-RFP (SY5Y/mito-RFP cells) and pmito-HyPer (SY5Y/mito-HyPer; mitochondrial hydrogen peroxide sensor) using Lipofectamine 2000, according to the manufacturer’s protocol (protocol #11668019, Thermo Fisher Scientific, Waltham, MA, USA). Transfectants were selected by growth in a medium containing 1 mg/mL of G418 (protocol #10131027, Thermo Fisher Scientific) for 7 days. After single-cell plating, stable clones were selected under a fluorescence microscope (IX71, Olympus, Tokyo, Japan). The cells were then cultured in Dulbecco’s modified Eagle’s medium that was supplemented with 10% fetal bovine serum and 1% penicillin–streptomycin (Invitrogen, Carlsbad, CA, USA).

### 2.2. Reagents and Gene Knockdown

The chemical reagents, including L-carnitine, acetyl-L-carnitine (Ac-L-Carnitine), MPP^+^, and Hoechst 33342 were purchased from Sigma-Aldrich (St. Louis, MO, USA) and Smoothened Agonist (SAG) was purchased from Calbiochem (San Diego, CA, USA). For the IFT88 knockdown experiment, the SH-SY5Y cells were transfected with a previously validated siRNA targeting *IFT88* (5′-CCGAAGCACUUAACACUUA-3′) and a scrambled siRNA (5′-CCUACGCCACCAAUUUCGU-3′) using Lipofectamine 2000 (#11668019, Thermo Fisher Scientific), according to the manufacturer’s protocol. The siRNAs were synthesized by Genolution (Seoul, Korea). At 48 h post-transfection, the cells were further treated with the indicated reagents.

### 2.3. Cilia Staining and Counting

To stain the primary cilia of the SH-SY5Y cells, they were fixed with 4% (*w*/*v*) paraformaldehyde (PFA) and 0.1% (*v*/*v*) Triton X-100. The cells were then blocked with phosphate-buffered saline (PBS) containing 1% bovine serum albumin (BSA) before incubation overnight at 4 °C with primary antibodies against ARL13B (17711-1-AP, 1:1000 Proteintech, Chicago, IL, USA). After washing, the cells were incubated with Alexa Fluor 488-conjugated secondary antibodies at room temperature for 1 h. The cells were then stained with Hoechst 33342 dye (H3570, 1:10,000, Thermo Fisher). The cilia images were acquired using a fluorescence microscope. The cilia counts were obtained for approximately 200 cells for each experimental condition (*n* = 3). The percentage of ciliary cells was calculated as follows: (total number of cilia/total number of nuclei in each image) × 100. The cilia lengths were measured using the free-hand line selection tool of the Olympus cellSens imaging software (Olympus Europa, Hamburg, Germany), and the average cilium length was calculated. The graphical analyses were performed using GraphPad Prism 8 (GraphPad Software, San Diego, CA, USA).

### 2.4. Evaluation of Mitochondrial Fragmentation and ROS Generation 

To evaluate mitochondrial fragmentation, the treated SH-SY5Y/mito-RFP cells were fixed with 4% PFA. Mitochondrial images were obtained using a fluorescence microscope (Cal Zeiss). The mitochondrial lengths were measured using the freehand line selection tool in CellSens Standard (Olympus Europa Holding GmbH, Hamburg, Germany). The mean length of the mitochondria was determined by measuring 20–30 linearized and unconnected filament-like mitochondria per cell (*n* = 3 independent experiments). We also obtained images of at least 10 randomly selected cells per individual, which were then digitized and analyzed using GraphPad Prism 8. The mitochondria-specific ROS levels were assessed using the HyPer protein system. The pHyPer-dMito vector encoding the mitochondria-targeted HyPer (Mito-HyPer) was obtained from Eyrogen (San Diego, CA, USA). The HyPer assay was performed by plating SY5Y/mito-HyPer cells and exposing them to treatment for 24 h. They were then imaged with fluoresce microscopy. The relative mitochondrial ROS ratio is presented here as the change in fluorescence of the drug-treated sample pictures compared to the control samples.

### 2.5. Western Blot Analysis

For western blotting, all lysates were prepared in 2 × Laemmli sample buffer (62.5 mM Tris-HCl, pH 6.8, 25% (*v*/*v*) glycerol, 2% (*w*/*v*) sodium dodecyl sulfate (SDS), 5% (*v*/*v*) β-mercaptoethanol, and 0.01% (*w*/*v*) bromophenol blue (Bio-Rad, Hercules, CA, USA)). All cellular proteins were quantified using Bradford solution (Bio-Rad), according to the manufacturer’s instructions. The samples were then separated using SDS-polyacrylamide gel electrophoresis and transferred onto a polyvinylidene fluoride membrane (Bio-Rad). After blocking with 4% (*w*/*v*) skim milk in Tris-buffered saline plus Tween (TBST, 25-mM Tris, 140-mM sodium chloride, and 0.05% (*v*/*v*) Tween^®^ 20), the membranes were incubated overnight with primary antibodies that were specific to the following target proteins: actin (MAB1501, 1:10,000 Millipore, MA, USA), IFT88 (13967-1-AP, 1:3000 Proteintech, Chicago, IL, USA), GLI2 (18989-1-AP, 1:3000 Proteintech), cleaved caspase-3 (#9661S, 1:2000 Cell Signaling Technology, MA, USA), phospho-NF-κB p65 (#3031, 1:1000 Cell Signaling Technology), and nuclear factor-κB (NF-κB) p65 (#8242, 1:2000 Cell Signaling Technology). For protein detection, the membranes were incubated with horseradish peroxidase (HRP)-conjugated secondary antibodies (Cell Signaling Technology). 

### 2.6. CCK8 and ELISA 

The cell viability was determined using a Cell Counting Kit-8 assay kit (CCK-8, Dojindo, Rockville, MD, USA), and all procedures were performed according to the manufacturer’s instructions. The cytokine levels in the cell culture supernatants were determined using an enzyme-linked immunosorbent assay (ELISA). The levels of human IL-6 and TNF-α in the cell culture supernatants were separately determined using ELISA kits from BD Biosciences (San Jose, CA, USA) and Abcam (Cambridge, MA, USA). All procedures were performed according to the manufacturer’s instructions. 

### 2.7. Statistical Analysis

The data were obtained from at least three independent experiments and are presented as mean ± standard error of the mean (SEM). The statistical evaluation of the results was performed using one-way analysis of variance (*n* = 3, * *p* < 0.05, ** *p* < 0.01, *** *p* < 0.001, n.s. = not significant). 

## 3. Results

### 3.1. Carnitine Promotes Primary Ciliogenesis in SH-SY5Y Cells 

The primary cilia control various cellular processes, including cell signaling and development. In order to identify novel metabolites that induce primary ciliogenesis, we previously developed a cell-based screening system using RPE cells. By using this screening system, we have identified several candidates for regulating ciliogenesis by screening a library of fecal metabolites [15]. Among these candidates, we have identified both L-carnitine and acetyl-L-carnitine as inducers of primary ciliogenesis. However, the role of carnitine in primary ciliogenesis remains unclear. In order to confirm our screening results, we investigated Smoothened (Smo) protein recruitment to cilia in RPE/Smo-GFP cells after treatment with either SAG, which is an agonist for the protein Smo, or L-carnitine and acetyl-L-carnitine. As expected from the screening results, treatment with either L-carnitine or acetyl-L-carnitine strongly increased the recruitment of the Smo-GFP protein in REP cells (Figure 1A,B). In addition, we found that treatment with carnitines increased the expression of GLI2 in RPE cells (Figure 1C). Since both increased the primary ciliogenesis, and the recruitment of Smo can result in Smo-GFP-positive cells, we further examined the effect of carnitines on primary ciliogenesis in SH-SY5Y neuroblastoma cells. Because IFT88 regulates anterograde IFT, the loss of IFT88 blocks primary ciliogenesis by disrupting cilia assembly. We found that increased primary ciliogenesis was efficiently blocked by the depletion of IFT88 in SH-SY5Y by staining with a ciliary protein ARL13B (Figure 1E,F). These results implicated that carnitine is a strong inducer of primary ciliogenesis in different cell types.

### 3.2. Carnitine Inhibits MPP^+^-Mediated Mitochondrial ROS and Fragmentation in SH-SY5Y Cells 

The mitochondrial toxins, such as MPTP and its active derivative MPP^+^, cause syndromes that mimic PD because they induce the disruption of the mitochondrial membrane potential [12]. Dysfunctional and fragmented mitochondria produce higher levels of mitochondrial-derived ROS (mtROS), which contributes to neuronal toxicity. Carnitine has been reported to possess an antioxidant effect that prevents mitochondrial damage in a cellular PD model [16]. Therefore, we further investigated the effects of carnitine on mitochondrial ROS production. The measurement of mtROS, which was performed via evaluating the expression of a mitochondrial hydrogen peroxide sensor (mito-HyPer), has revealed that mtROS was excessively generated in MPP^+^-treated SH-SY5Y cells. However, the excess mtROS production that was induced by MPP^+^ was almost abrogated by a combination treatment containing L-carnitine or acetyl-L-carnitine and Mito-Q, which is an mtROS scavenger (Figure 2A,B). 

We recently reported that enhancing primary ciliogenesis reduces the mitochondrial stress in SH-SY5Y cells [14]. Thus, we explored the effects of carnitine on ciliogenesis in MPP^+^-treated SH-SY5Y cells. MPP^+^ induces mitochondrial fragmentation in SH-SY5Y cells (Figure 2C–E). However, co-treatment with L-carnitine or acetyl-L-carnitine suppressed the MPP^+^-induced mitochondrial fission. Notably, in the MPP^+^-treated SH-SY5Y cells, the loss of primary cilia following IFT88 depletion was accompanied by completely fragmented mitochondria (Figure 2C–E). These results suggest that carnitine induced the increases in primary ciliogenesis, which inhibits MPP^+^-induced mitochondrial fragmentation. 

### 3.3. Carnitine Inhibits MPP^+^-Induced Neurotoxicity by Enhancing Primary Ciliogenesis in SH-SY5Y Cells

Since carnitine has been shown to prevent neurotoxicity in an MPTP-induced PD model [17], SH-SY5Y cells were treated with MPP^+^ and with or without L-carnitine and acetyl-L-carnitine. The treatment of the SH-SY5Y cells with either L-carnitine or acetyl-L-carnitine significantly reduced MPP^+^-induced cell death and caspase-3 activation (Figure 3A,B). Next, we investigated the potential role of carnitine-mediated ciliogenesis in the protection against MPP^+^-induced neurotoxicity. The loss of primary cilia that was caused by the depletion of IFT88 abolished the neuroprotective effects of L-carnitine and acetyl-L-carnitine against MPP^+^-induced neurotoxicity in the SH-SY5Y cells (Figure 3C). Moreover, the inhibition of caspase-3 activation by L-carnitine and acetyl-L-carnitine in the MPP^+^-treated cells was lost when IFT88 was depleted (Figure 3D). 

### 3.4. Carnitine Inhibits the MPP^+^-Induced Inflammatory Response by Enhancing Primary Ciliogenesis in SH-SY5Y Cells

Exposure to MPTP exacerbates PD pathogenesis by increasing neural inflammation [18]. Moreover, we previously showed that 2-isopropylmalic acid (2-IPMA) suppresses the inflammatory response by enhancing primary ciliogenesis [15]. Thus, we next investigated the role of carnitine in the MPP^+^-mediated inflammatory response by measuring the levels of pro-inflammatory cytokines, such as IL-6 and TNF-α. As shown in Figure 4A,B, the MPP^+^ treatment consistently increased the expression levels of IL-6 and TNF-α in the SH-SY5Y cells, but this effect was significantly reduced by treatment with L-carnitine or acetyl-L-carnitine. In addition, we investigated the effect of primary ciliogenesis on MPP^+^-mediated inflammatory responses. Notably, the reduction in pro-inflammatory cytokines that was caused by L-carnitine or acetyl-L-carnitine was substantially restored following the loss of primary cilia via IFT88 depletion in the MPP^+^-treated cells (Figure 4A,B). Because NF-κB is a key inflammatory transcription factor, we further examined the effect of carnitine on the activation of NF-κB in the MPP^+^-treated cells. Consistently, both L-carnitine and acetyl-L-carnitine inhibited the activation of NF-κB by MPP^+^ in the SH-SY5Y cells (Figure 4C). In addition, the depletion of IFT88 further blocked the inhibitory effect of L-carnitine and acetyl-L-carnitine on NF-κB activation in MPP^+^-treated cells (Figure 4C). Taken together, these results suggest that carnitines suppress the inflammatory response in MPP^+^-treated SH-SY5Y cells. 

## 4. Discussion

The mechanism by which L-carnitine and acetyl-L-carnitine regulate primary ciliogenesis remains unknown. In this study, we have evaluated the potency of L-carnitine and acetyl-L-carnitine as inducers of primary cilia formation. L-carnitine is naturally found in most mammalian tissues. The primary sources of dietary carnitine are red meat and dairy products. L-carnitine and acetyl-L-carnitine—a derivative containing two carbons in the acyl moiety—are transported into cells via the novel organic cation transporter 2 [19]. L-carnitine is essential for a series of reactions, collectively referred to as the “carnitine shuttle”, which transfers the activated long-chain fatty acids into the mitochondria and facilitates fatty acid β-oxidation. The carnitine shuttle is also essential for preventing the accumulation of long-chain fatty acids and acyl-CoAs, which can be deleterious to cells [20,21]. Because acetyl-L-carnitine is metabolized to acetyl-CoA, it can also acetylate histones, which can modify gene expression, as well as acetylate proteins and enzymes [20,21]. In contrast, histone deacetylases (HDACs) control gene expression by removing acetyl groups from the lysine residues of the histone proteins, resulting in the epigenetic modulation of gene expression [22,23]. Thus, HDACs play a pivotal role in maintaining metabolic homeostasis. Several coenzyme A (CoA) derivatives, including acetyl-CoA, butyryl-CoA, HMG-CoA, and malonyl-CoA, act as allosteric activators of HDACs [21,24]. Moreover, several HDACs, including HDAC2, HDAC3, HDAC6, and HDAC8, control primary cilia assembly and elongation in various tissues [25,26,27,28]. In this study, we found that treatment with either L-carnitine or acetyl-L-carnitine efficiently promoted primary ciliogenesis in different cells (Figure 1). Thus, we expect that further investigations into the role of HDACs in acetyl-L-carnitine-mediated primary ciliogenesis may be helpful in elucidating the underlying mechanism. 

Recently, there has been increasing interest in the therapeutic potential of L-carnitine and acetyl-L-carnitine for neuroprotection [20,29]. It has been reported that the administration of acetyl-L-carnitine after injury can improve mitochondrial function, decrease swelling in the brain, and prevent tissue loss in pediatric injury models [20,29,30,31]. In addition, the long-term administration of acetyl-L-carnitine can improve the brain energy status in healthy mice [29,32]. Moreover, carnitine was present at elevated levels in plasma samples that were obtained from a group of PD patients [33]. 

We found that treatment with L-carnitine and acetyl-L-carnitine decreased mitochondrial fragmentation and ROS production in the MPP^+^-treated cells (Figure 2). Moreover, MPP^+^-induced neurotoxicity was reduced after treatment with L-carnitine and acetyl-L-carnitine, which promoted primary ciliogenesis in the SH-SY5Y cells (Figure 3). The primary cilia contribute to maintaining homeostasis by regulating many cellular processes. Recently, we reported that primary cilia decrease oxidative stress in order to protect neuronal cells [14]. Moreover, the blockade of primary ciliogenesis potentiated neuronal loss and motor disability in a PD model [14]. In addition, primary cilia can enhance retinal ganglion cell survival after axotomy [34], indicating that primary cilia confer protective effects by reducing oxidative stress. Furthermore, the dysregulation of primary cilia has been linked to neurodegenerative diseases other than PD. A lack of huntingtin (HTT) dysregulation results in reduced and aberrant primary ciliogenesis, whereas mutants with increased HTT levels show increased ciliogenesis [35]. In addition, the targeted depletion of primary cilia was observed in dopaminergic neurons in a mouse model of Huntington’s disease (HD) [36]. In this study, we found that L-carnitine and acetyl-L-carnitine reduced the mitochondrial ROS production and cell death in the SH-SY5Y cells (Figure 3). The protective effects of L-carnitine and acetyl-L-carnitine against neuronal toxicity have been established, but the mechanistic relationship between L-carnitine and/or acetyl-L-carnitine and primary ciliogenesis has not yet been elucidated. Here, we conclude that increased primary ciliogenesis that is induced by L-carnitine and acetyl-L-carnitine contributes to a neuronal protective effect against MPP^+^-induced oxidative stress and cell death. Our results also indicate that the inhibition of primary cilia that is caused by IFT88 depletion completely reversed the protective effect of L-carnitine and acetyl-L-carnitine against MPP^+^-induced cell death (Figure 3). Furthermore, we determined that the anti-inflammatory effects of L-carnitine and acetyl-L-carnitine act against MPP^+^-induced cell stress by promoting primary ciliogenesis in SH-SY5Y cells (Figure 4). This is consistent with our previous reports, wherein we showed that 2-IPMA ameliorates PM2.5-induced inflammation by promoting primary ciliogenesis in RPE cells [15]. Therefore, future investigations that are aimed at identifying the mechanisms underlying primary ciliogenesis will be helpful in understanding the neuroprotective effects of L-carnitine and acetyl-L-carnitine in neurodegenerative diseases.

## 5. Conclusions

L-carnitine and acetyl-L-carnitine are strong inducers of primary ciliogenesis and ameliorate MPP^+^-induced mitochondrial dysfunction and cell death by promoting primary ciliogenesis in SH-SY5Y cells. Although additional mechanistic and preclinical experiments are required, our findings suggest that L-carnitine and acetyl-L-carnitine are potential therapeutic agents for neurodegenerative diseases that are associated with the loss of primary cilia.

## Figures and Tables

**Figure 1 cells-11-02722-f001:**
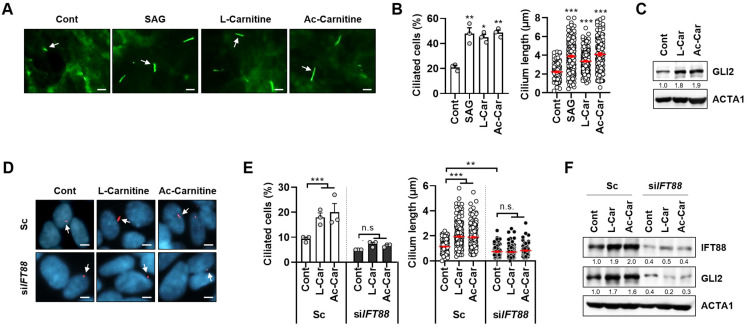
Carnitines stimulate primary ciliogenesis in RPE and SH-SY5Y cells. (**A**–**C**), RPE cells stably expressing smo-GFP protein (RPE/Smo-GFP) were treated with Smoothened Agonist (SAG) (1 µM), L-carnitine (L-Car, 100 µM), or acetyl-L-carnitine (Ac-Car, 100 µM) for 24 h. Then, the Smo-GFP protein was imaged using fluorescence microscopy. White arrows indicate Smo-GFP (A). The ciliary cells and cilium lengths were measured using a fluorescence microscope (**B**). The RPE/Smo-GFP cells were harvested and analyzed by western blotting with indicated antibodies (**C**). (**D**–**F**), SH-SY5Y cells transiently transfected with either scrambled siRNA (Sc) or targeted siRNA for *IFT88* (si*IFT88*) were treated with L-carnitine (100 µM) or Ac-L-carnitine (100 µM) for 24 h. Cells were stained with ARL13B antibody (red) and Hoechst dye (blue) (**D**). The ciliary cells and cilium lengths were measured (**E**). The cells were harvested and analyzed by western blotting with indicated antibodies (**F**). Data are presented as the mean ± SEM (*n* = 3, * *p* < 0.05, ** *p* < 0.01, *** *p* < 0.001, n.s. = not significant). Scale bar: 5 µm.

**Figure 2 cells-11-02722-f002:**
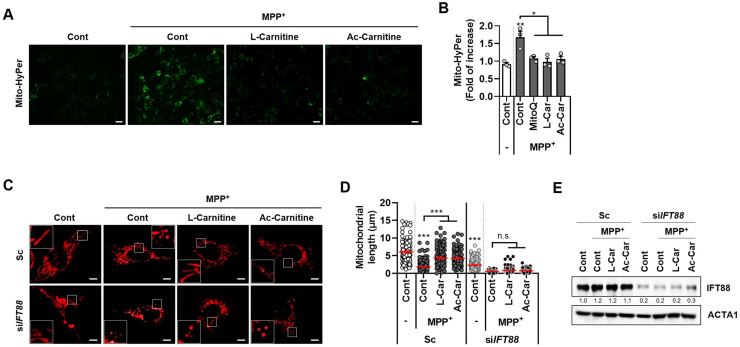
Carnitine inhibits excessive generation of mitochondrial ROS and mitochondrial fragmentation by MPP^+^ in SH-SY5Y cells. (**A**,**B**), SH-SY5Y cells stably expressing mito-HyPer protein (SY5Y/mito-HyPer cells) were treated with MPP^+^ (2.5 mM) in the presence or absence of L-carnitine (L-Car, 100 µM) or Ac-L-carnitine (Ac-Car, 100 µM) for 24 h. Next, the fluorescence intensity of mito-HyPer was imaged (**A**) and the fluorescence was measured (B). Scale bar: 20 µm. Data are presented as the mean ± SEM (*n* = 3, * *p* < 0.05, ** *p* < 0.01). (**C**–**E**), SH-SY5Y cells stably expressing mito-RFP protein (SY5Y/mito-RFP cells) were transiently transfected with either scrambled siRNA (Sc) or targeted siRNA for *IFT88* (si*IFT88*). These cells were then treated with MPP^+^ (2.5 mM) in the presence or absence of L-Carnitine (100 µM) or Ac-L-Carnitine (100 µM) for 24 h. Then, Mito-RFP in the cells was imaged using fluorescence microscopy (**C**), and the mitochondrial length was assessed (**D**). The SH-SY5Y cells were harvested and analyzed by western blotting with indicated antibodies (**E**). Data are presented as the mean ± SEM (*n* = 3, *** *p* < 0.001, n.s. = not significant). Scale bar: 5 µm.

**Figure 3 cells-11-02722-f003:**
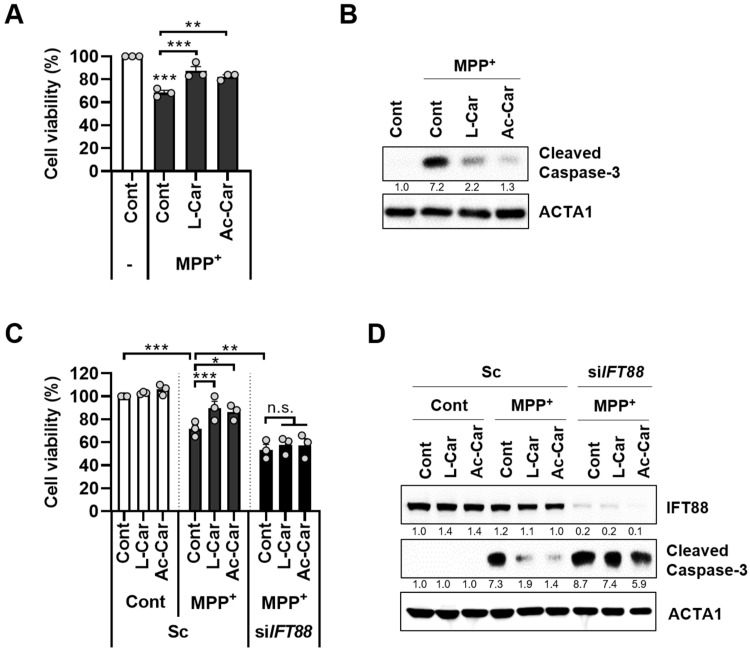
Carnitine inhibits MPP^+^-induced neurotoxicity by enhancing ciliogenesis in SH-SY5Y cells. (**A**,**B**), SH-SY5Y cells were treated with MPP^+^ (2.5 mM) in the presence or absence of L-carnitine (L-Car, 100 µM) or Ac-L-carnitine (Ac-Car, 100 µM) for 24 h. Then, cell viability was measured by a CCK-8 assay (A). The cells were further analyzed by western blotting with cleaved caspase-3 antibody (**B**). (**C**,**D**), SH-SY5Y cells were transiently transfected with either scrambled siRNA (Sc) or targeted siRNA for *IFT88* (si*IFT88*). The cells were then treated with MPP^+^ (2.5 mM) in the presence or absence of L-carnitine (100 µM) or Ac-L-carnitine (100 µM) for an additional 24 h. Cell death was measured using a CCK-8 assay (**C**), and western blotting was performed with indicated antibodies (**D**). Data are presented as the mean ± SEM (*n* = 3, * *p* < 0.05, ** *p* < 0.01, *** *p* < 0.001, n.s. = not significant).

**Figure 4 cells-11-02722-f004:**
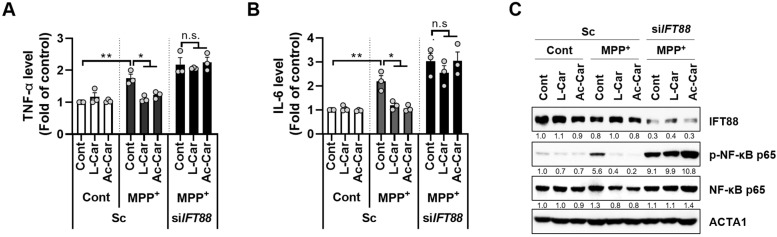
Carnitine inhibits the production of pro-inflammatory cytokines in MPP^+^-treated SH-SY5Y cells**.** SH-SY5Y cells transfected with either scrambled siRNA (Sc) or siRNA against *IFT88* (si*IFT88*) were treated with MPP^+^ (2.5 mM) in the presence or absence of L-carnitine (L-Car, 100 µM) or Ac-L-carnitine (Ac-Car, 100 µM) for 24 h. The expression levels of TNF-α (**A**) and IL-6 (**B**) under the relevant culture conditions were measured using an ELISA kit. The treated cells were further analyzed by western blotting using indicated antibodies (**C**). Data are presented as the mean ± SEM (*n* = 3, * *p* < 0.05, ** *p* < 0.01, n.s. = not significant).

## Data Availability

Not applicable.

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
