# Peer review of "Carnitine Protects against MPP+-Induced Neurotoxicity and Inflammation by Promoting Primary Ciliogenesis in SH-SY5Y Cells"

_cells, 2022, doi:10.3390/cells11172722_

Round 1

Reviewer 1 Report

The authors provide evidence that carnitine provides neuroprotection by enhancing the presence of primary cilia. The findings are of interest to both the cilia field and the neurodegeneration field. Overall, the paper is clearly written, but the explanation, presentation and interpretation of the data should be strengthened to ensure the claims of the paper reflect the data. Detailed comments are below:

1.       Major: The screening strategy employed in this paper and in reference 15 from the same group is difficult to interpret. Increased presence of Smo + cilia could be due to increased ciliogenesis or it could be due to increased Smo recruitment into cilia. Without simultaneous labeling of both Smo and a structural cilia marker, the recruitment versus ciliogensis scenarios can’t be teased apart. This needs to be clarified to properly interpret the results of this study.

2.       Major: The statistics were conducted with an n=3 but the scatter plots show more than 3 data points in all figures. This needs to be explained.

3.       Major: The Mito-HyPer experiments are difficult to interpret due to low resolution images in Figure 2. These images appear to have multiple layers of text data overlaid in the bottom left corner. These images, in their current form, are not publication quality. Moreover, the legend says that fluorescence was measured in two different ways (plate reader or microscopy). The data analyzed for these experiments should all be collected using the same technique in order to make valid statistical comparisons.

4.       Major: The mitochondria images in Figure 3 are difficult to interpret. It is not clear how mitochondria morphology was analyzed from these images to produce unambiguous measurements of mitochondria length. The mitochondria analysis approach needs to be clarified to make sense of the quantitative mitochondria analysis data.

5.       Major: The authors introduce reference 14 but then state that the relationship between primary cilia and mitochondrial dysfunction in neurodegeneration is not clear. The entire focus of reference 14 is establishing a relationship between cilia, mitochondrial stress, and PD. The major contributions of the current manuscript and how it relates to reference 14 needs to be clearly explained.

6.       Major: The authors state that “carnitine restores ciliary dysgenesis caused by MPP by inhibiting the inflammatory response in SH-SY5Y cells.” This seems to be an overstatement. The data indicate that carnitine may increase the presence of cilia and that it may reduce the inflammatory cascade. However, the data do not provide solid evidence that carnitine’s effects on cilia presence are mediated by the inflammatory cascade.

7.       There are grammar issues that should be resolved. The term “ciliate” is used incorrectly in the manuscript. The authors refer to “previous reports” when referring to previous figures, which is confusing (see section 3.3 for an example). Stating that carnitine restores ciliary dysgenesis is hard to follow.

8.       The description of ciliogenesis factors in the introduction is a bit arbitrary. The focus on a few of the many factors involved in ciliogenesis is confusing (Arl13b, Smo, and IFT88 compared to the many other critical factors).

9.       Blue nuclei and green cilia are hard visualized as merged images. For example, see figure 1C.

Reviewer 2 Report

The manuscript is concisely written with adequate literature cited. As a suggestion, I would refrain from using word implicating higher significance than is shown as e.g. in "We found that both L-carnitine and acetyl-L-carnitine strongly promoted primary ciliogenesis in SH-SY5Y dopaminergic neuroblastoma cells." I would shorten teh result section by text editing, it is too lengthy.

 Also I would consider joining figs. 2+3 and 4+5 as they contain single panel of ICF and graph or WB panel and graph and are showing data that can be easily put together as they present closely related data. also some graphic editing should be done to avoid having too much free space around the figures (e.g. fig4). 

Fig3 must be improved by adding ICF panel to 3A from the IFT88 knockdown (shown in graph in 3B) that is missing and is necessary. 

In fig3, as i dont see any visually clear differences between the MPP-/cont, MPP+/cont condition and consequently any other condition, authors must show what is exactly measured as e.g.insets of enlarged areas of the cells as i dont see where the differences shown in the 3B are coming from and have a hard time believing there are any differences at the moment. 

To sum up,, althought the data are interesting, the manuscript seems too preliminary and descriptive to my liking, the authors should show at least some fuctional data, e.g. SHH activation by treatment of cells by L-carnitine as assayed by SMO recruitment measured from ICF or RT-PCR of Gli/ptch target genes  which should not be too complicated. also if they show actions of L-carnitine on cilia length/numbers and SHH signaling in other established ciliary models (e.g. NIH3T3 cells ) it would strengthen the data.

Round 2

Reviewer 1 Report

Q1. The clarification in the text helps explain the interpretation of the screening strategy and the validation with Arl13b. However, there are typos and grammatical errors throughout the added text. This should be proofread. I appreciate and understand that writing in a non-native language is very challenging.

Q2. The author’s explanation is difficult to understand. The data for % of ciliated cells does not show scatter plots. The data for cilia length does show scatter plots (see Figure 1B as an example). I still do not understand how the cilium length data in Figure 1B has an n=3. This is still not explained in the methods. A clear description of the sample size is needed.

Q3. The images are improved. However, the figure legend has grammatical issues that make it difficult to understand.

Q4. I appreciate the clarification of the method and the improved images help substantially. It would be wonderful to show example filaments that are linearized. It seems that those images would be a more accurate representation of the data plotted in figure 2D than the images of the entire cell.

Q5. The description in the discussion is fine. However, it seems strange to cite a paper with a title “Primary cilia mediate mitochondrial stress responses to promote dopamine neuron survival in a Parkinson’s disease model” and then to follow that with a sentence that states “…there is little evidence of the relationship between primary cilia and mitochondrial dysfunction in neurodegenerative diseases.” I believe that sentence should be reworded or omitted because the group’s prior work did establish a relationship between primary cilia and mitochondrial dysfunction in neurodegenerative disease.

Q6. Great.

Q7. The term “ciliate” generally refers to single cell organisms that contain cilia (usually protists). Ciliated cell is fine, but the term ciliate is still used in the introduction in line 143. I believe this should be “ciliary” rather than “ciliate”.

Q8. Great.

Q9. Great.

Author Response

Please check attached file

Q1. The clarification in the text helps explain the interpretation of the screening strategy and the validation with Arl13b. However, there are typos and grammatical errors throughout the added text. This should be proofread. I appreciate and understand that writing in a non-native language is very challenging.

Response 1:  We appreciate your kind comments. We further corrected some part to improve this manuscript.  In addition, this manuscript has been edited by a professional English editing service (Editage; editage.co.kr). I attached a certification letter for this manuscript by Editage.

Q2. The author’s explanation is difficult to understand. The data for % of ciliated cells does not show scatter plots. The data for cilia length does show scatter plots (see Figure 1B as an example). I still do not understand how the cilium length data in Figure 1B has an n=3. This is still not explained in the methods. A clear description of the sample size is needed.

Response 2:  We appreciate your comment.  According to the reviewer’s comment, we have further corrected the graphs and legends to show scatter plots.  

Q3. The images are improved. However, the figure legend has grammatical issues that make it difficult to understand.

Response 3: According to the reviewer’s comment, we have corrected the figure legend again.

Q4.   I appreciate the clarification of the method and the improved images help substantially. It would be wonderful to show example filaments that are linearized. It seems that those images would be a more accurate representation of the data plotted in figure 2D than the images of the entire cell.

Response 4:  We appreciate the comment to improve this manuscript.  According to the reviewer’s suggestion, we have further corrected the mitochondrial images.  

Q5.  The description in the discussion is fine. However, it seems strange to cite a paper with a title “Primary cilia mediate mitochondrial stress responses to promote dopamine neuron survival in a Parkinson’s disease model” and then to follow that with a sentence that states “…there is little evidence of the relationship between primary cilia and mitochondrial dysfunction in neurodegenerative diseases.” I believe that sentence should be reworded or omitted because the group’s prior work did establish a relationship between primary cilia and mitochondrial dysfunction in neurodegenerative disease.

Response 5: Thank you for your valuable comments According to the reviewer’s suggestion, we have omitted the sentence.  

Q6.  The term “ciliate” generally refers to single cell organisms that contain cilia (usually protists). Ciliated cell is fine, but the term ciliate is still used in the introduction in line 143. I believe this should be “ciliary” rather than “ciliate”.

Response 6: We appreciate your suggestion. I agree to your comment. According to this generous suggestion, we have corrected the sentence to ‘ciliary’ rather than ‘cilated’
